# Characterization of Copper(II) and Zinc(II) Complexes of Peptides Mimicking the CuZnSOD Enzyme

**DOI:** 10.3390/molecules29040795

**Published:** 2024-02-08

**Authors:** Enikő Székely, Mariann Molnár, Norbert Lihi, Katalin Várnagy

**Affiliations:** Department of Inorganic and Analytical Chemistry, University of Debrecen, Egyetem Square 1, H-4032 Debrecen, Hungary

**Keywords:** multihistidine peptide, Cu(II)-complex, Zn(II)-complex, SOD activity, CuZnSOD

## Abstract

Antimicrobial peptides are short cationic peptides that are present on biological surfaces susceptible to infection, and they play an important role in innate immunity. These peptides, like other compounds with antimicrobial activity, often have significant superoxide dismutase (SOD) activity. One direction of our research is the characterization of peptides modeling the CuZnSOD enzyme and the determination of their biological activity, and these results may contribute to the development of novel antimicrobial peptides. In the framework of this research, we have synthesized 10, 15, and 16-membered model peptides containing the amino acid sequence corresponding to the Cu(II) and Zn(II) binding sites of the CuZnSOD enzyme, namely the Zn(II)-binding HVGD sequence (80–83. fragments), the Cu(II)-binding sequence HVH (fragments 46–48), and the histidine (His63), which links the two metal ions as an imidazolate bridge: Ac-FHVHEGPHFN-NH_2_ (L^1^(10)), Ac-FHVHAGPHFNGGHVG-NH_2_ (L^2^(15)), and Ac-FHVHEGPHFNGGHVGD-NH_2_ (L^3^(16)). pH-potentiometric, UV-Vis-, and CD-spectroscopy studies of the Cu(II), Zn(II), and Cu(II)-Zn(II) mixed complexes of these peptides were performed, and the SOD activity of the complexes was determined. The binding sites preferred by Cu(II) and Zn(II) were identified by means of CD-spectroscopy. From the results obtained for these systems, it can be concluded that in equimolar solution, the –(NGG)HVGD- sequence of the peptides is the preferred binding site for copper(II) ion. However, in the presence of both metal ions, according to the native enzyme, the -HVGD- sequence offers the main binding site for Zn(II), while the majority of Cu(II) binds to the -FHVH- sequence. Based on the SOD activity assays, complexes of the 15- and 16-membered peptide have a significant SOD activity. Although this activity is smaller than that of the native CuZnSOD enzyme, the complexes showed better performance in the degradation of superoxide anion than other SOD mimics. Thus, the incorporation of specific amino acid sequences mimicking the CuZnSOD enzyme increases the efficiency of model systems in the catalytic decomposition of superoxide anion.

## 1. Introduction

The superoxide anion, O_2_^−^, is produced by the one-electron reduction of dioxygen in numerous biochemically relevant redox processes [1]. These very toxic species are thought to play a role in diseases and pathological processes, such as aging, cancer, or membrane or DNA damage, etc. [2]. Cells are protected against this species by superoxide dismutase (SOD)—a metalloenzyme, which very efficiently catalyzes the dismutation of superoxide anion into H_2_O_2_ and O_2_. With this reaction, SOD reduces the risk of oxidative stress by eliminating the highly reactive superoxide [3]. In addition, SOD1 mutations are associated with human familial amyotrophic lateral sclerosis (ALS) [4,5]. SODs are also expected to play a role in modulating innate immunity, as O_2_ is one of several toxic substances used by the cellular arm of the immune system to kill bacteria and virus-infected cells [6,7]. CuZnSODs are metalloproteins whose structure is provided by a zinc atom, and a copper atom is the catalytic cofactor [8]. In vivo, the zinc atom is likely to be transported to the enzyme by passive diffusion, while the copper atom is transported by a copper chaperone via a translocation mechanism in a process that requires the formation of a SOD-copper chaperone heterodimer [9,10,11,12,13].

Antimicrobial peptides are short cationic peptides that are present on biological surfaces susceptible to infection and play an important role in innate immunity [14]. A very large number of different antimicrobial peptides have been discovered. Many antimicrobial peptides are characterized by an amphipathic structure [15]. Antimicrobial peptides include, for example, linear peptides that may adopt an α-helix conformation upon binding to bacteria [16,17,18,19,20,21,22,23,24,25,26]. There are peptides that form β-sheets through cysteine coupling and cysteine-constrained loop structures [27,28,29]. However, there are also antimicrobial peptides that do not have such ordered structures but are characterized by an overrepresentation of certain amino acids [14].

Considering the role of the secondary structure, the amphiphilic nature of peptides, and the composition of bacterial membranes, it was thought that the function of antimicrobial peptides involves direct binding to the lipid bilayer and that interaction with bacterial membranes is a prerequisite for the function of antimicrobial peptides. However, the mechanism of the action of antimicrobial peptides on bacteria is complex [14,30,31,32]. On this basis, we distinguish between membrane-disrupting and non-membrane-disrupting peptides [14,32]. In several cases, heparin-binding motifs [33], as well as peptide sequences of endogenous proteins [34,35,36], matrix proteins [37], growth factors [38], and histidine-rich glycoproteins [39,40,41,42,43,44,45,46,47] have been shown to have antimicrobial activity, and are thus classical peptides and proteins of innate immunity. It is worth noting that there are other ways to prevent bacteria growth on surfaces, like polymeric coating or metal ions, on the surface [48,49]. These competitive systems are also important because the use of human-type peptides carries the risk of resistance.

These peptides, like other compounds [50,51,52] with antimicrobial activity, often have significant SOD activity. Based on these observations, the possible antimicrobial effects of the C-terminal heparin-binding domain of CuZnSOD were also investigated [15]. The results show that the CuZnSOD peptide segment has antimicrobial activity, which also means that peptides of human proteins may help to develop antimicrobial peptides based on endogenous peptides.

One direction of our research is the characterization of peptides modeling the CuZnSOD enzyme and the determination of their biological activity [53], and these results may contribute to the development of novel antimicrobial peptides.

The two binding sites of the CuZnSOD enzyme are the Zn(II)-binding HVGD sequence (fragments 80–83) and the Cu(II)-binding HVH sequence (fragments 46–48), and the two metal ions are connected by the imidazolatee ring of His63 [5]. In order to mimic the two binding sites, we synthesized 10-, 15-, and 16-membered peptides containing two or three binding sites in an environment corresponding to the native enzyme: Ac-FHVHEGPHFN-NH_2_ (L^1^(10)), Ac-FHVHAGPHFNGGHVG-NH_2_ (L^2^(15)), and Ac-FHVHEGPHFNGGHVGD-NH_2_ (L^3^(16)). pH-potentiometric, UV-Vis-, and CD-spectroscopy studies of the Cu(II), Zn(II), and Cu(II)-Zn(II) mixed complexes of these peptides were performed, and the SOD activity of the complexes was tested.

## 2. Experimental

### 2.1. Chemicals

To synthesize the oligopeptides (Ac-FHVHEGPHFN-NH_2_ (L^1^(10)), Ac-FHVHAGPHFNGGHVG-NH_2_ (L^2^(15)), and Ac-FHVHEGPHFNGGHVGD-NH_2_ (L^3^(16))), solid phase peptide synthesis was performed using a microwave-assisted Liberty 1 Peptide Synthesizer (CEM, Matthews, NC, USA). Fmoc/tBtu technique and TBTU/HOBt/DIPEA activation strategy were used. A detailed description of the procedure has already been published in our previous papers [54,55,56]. The chemicals and solvents used for synthetic purposes were obtained from commercial sources in the highest available purity and used without further purification. The Rink Amide AM resin (substitution: 0.70 mmole/eq), all of the N-fluorenylmethoxycarbonyl (Fmoc)-protected amino acids (Fmoc-Ala-OH, Fmoc-Asp(O*t*Bu)-OH (O*t*Bu: 5-tert-butyl), Fmoc-Phe-OH, Fmoc-Gly-OH, Fmoc-His(Trt)-OH (TrT: trityl), Fmoc-Val-OH Fmoc-Glu(O*t*Bu)-OH), and 2-(1-H-benzotriazole-1-yl)-1,1,3,3-tetramethyluronium tetrafluoroborate (TBTU) are Novabiochem (Switzerland) products. N-hydroxybenzotriazole (HOBt·H_2_O), N-methylpyrrolidone (NMP), triisopropylsilane (TIS), 2,2′-(ethylenedioxy)diethanethiol (DODT), and 2-methyl-2-butanol were purchased from Sigma-Aldrich Co., St. Louis, MO, USA, while N,N-diisopropyl-ethylamine (DIPEA) and trifluoroacetic acid (TFA) were Merck Millipore Co. products, Burlington, MA, USA. Peptide-synthesis grade N,N-dimethylformamide (DMF) and acetic anhydride (Ac_2_O) were bought from VWR International, while piperidine, dichloromethane (DCM), diethyl ether (Et_2_O), acetic acid (96%) (AcOH), and acetonitrile (ACN) were from Molar Chemicals Ltd., Halasztelek, Hungary. The peptide was characterized by ESI-MS, and the data is collected in Table A1.

The concentrations of the peptide stock solutions were determined by pH-potentiometric titrations. The stock solutions of copper(II) chloride, nickel(II) chloride, and zinc(II) chloride were prepared from analytical grade reagents, and their concentrations were checked gravimetrically via the precipitation of oxinate.

### 2.2. High-Performance Liquid Chromatography

The purity of the synthesized product was checked by analytical RP-HPLC monitoring the absorbance at 222 nm using a Jasco instrument equipped with a Jasco MD-2010 plus multiwavelength detector. The elution method was set as 0% of solvent B at 0 min, which begins to increase after 1 min up to 12% in 14 min and decreases to 0% again after 9 min. The gradient profile was achieved using solvent A (0.1 *v*/*v*% TFA in water) and solvent B (0.1 *v*/*v*% TFA in MeCN) at a flow rate of 1.0 mL/min. The solid phase was a Teknokroma Europa Protein C18 chromatographic column (250 mm × 4.6 mm, 300 Å pore size, 5 μm particle size) in the separation procedure.

The peptides were separated by RP-HPLC semipreparative Teknokroma Europa Protein 300 C18 column (250 mm × 10 mm, 5 μm). Reverse phase HPLC was performed on a Jasco instrument equipped with a Jasco UV-2077 Plus 4-λ Intelligent UV/Vis detector. The flow rate of 3 mL·min^−1^ was maintained. The elution of peptides was monitored by UV absorbance at 222 nm. The gradient elution mentioned above was used in this system.

### 2.3. Potentiometric Measurements

In total, 3 mL aliquots of the ligands and 0.2 M carbonate-free potassium hydroxide solution titrant were used for pH-potentiometric measurements. In order to avoid side reactions with carbon dioxide and/or oxygen, the headspace over the sample was purged with argon gas during titration, in which the concentration of the ligand was ca. 2 mM. KCl in 0.2 M concentration was used as a background electrolyte. Interaction with metal ions was studied in samples containing copper(II) chloride, zinc(II) chloride, or nickel(II) chloride at 1:3, 1:1, and 2:1 metal-to-ligand ratios. All pH-potentiometric measurements were carried out at 298 K, and an IKA Topolino magnetic stirrer, IKA, Wilmington, NC, USA was used to stir the samples. To perform the titrations, a MOL-ACS microburette was used, controlled by a computer, and a Molspin pH-meter equipped with a Metrohm 6.0234.100 combination glass electrode detected the pH data which were converted into hydrogen ion concentrations according to the method described by Irving et al. [57] SUPERQUAD [58] and PSEQUAD [59] computational programs enabled the calculation of the protonation constants of the ligands and the stability constants (log*β_pqr_* for M*_p_*H*_q_*L*_r_*species) of the metal complexes. Equations (1) and (2) define the equilibrium constants:(1)pM+qH+rL⇌MpHqLr
(2)$$\beta_{pqr}=\frac{\left[M_{p}H_{q}L_{r}\right]}{{\left[M\right]}^{p}\cdot {\left[H\right]}^{q}\cdot {\left[L\right]}^{r}}$$

### 2.4. Spectroscopic Studies

UV-visible absorption spectra were recorded under the same conditions as the pH-potentiometric measurements in a 2.5 mL solution at different pH values. A Perkin Elmer Lambda 25 spectrophotometer, PerkinElmer, Waltham, MA, USA was used, and the absorption spectra were recorded in the 250–1100 nm wavelength range for copper(II) and nickel(II) containing systems at a metal-to-ligand ratio of 1:2 to 2:1 in a 1.00 cm cuvette.

The circular dichroism (CD) spectroscopic measurements were carried out on a Jasco-810 spectropolarimeter using the same ligand and metal concentrations and ratios as described above. CD-spectra were recorded in silica cells of 1.00 cm path length in the 280–800 nm range and 0.10 cm between 220 and 300 nm.

### 2.5. Superoxide Dismutase Activity

The superoxide dismutase (SOD) activity of the complexes was determined by the indirect method of NBT reduction [60]. The superoxide anion was generated in situ by the xanthine/xanthine oxidase reaction and detected spectrophotometrically by monitoring the reduction of NBT at 560 nm. The tests were carried out in phosphate buffer (50 mM) at pH 6.8 containing NBT (4.5 × 10^−2^ mM) and xanthine (0.2 mM). The reaction was initiated by adding an appropriate amount of xanthine oxidase to generate a change in absorbance around ΔA560 = 0.020–0.025 min^−1^. The NBT reduction rate was measured in the presence and the absence of the investigated system ([Cu(II)]tot = 0–2.1 µM) for 480 s. For the greater reproducibility of the measured data, we started each measurement by monitoring a blank sample (without any Cu(II) complex) for 3–4 min, then the Cu(II) complex—prepared in phosphate buffer—was added to the sample. The change in the absorbance was monitored for another 4 min. The corresponding rates were obtained by fitting the experimental data to a straight line. The SOD activity was then expressed by the IC_50_ values (the concentration that causes 50% inhibition of NBT reduction). For calculating the inhibition, the following equation is used:(3)$$inhibition\%=\frac{{\left(\frac{\Delta A}{time}\right)}^{blank}-{\left(\frac{\Delta A}{time}\right)}^{complex}}{{\left(\frac{\Delta A}{time}\right)}^{blank}}\times 100$$
where ${\left(\frac{\Delta A}{time}\right)}^{blank}$ is the change in absorbance per minute at 560 nm in the absence of complex and ${\left(\frac{\Delta A}{time}\right)}^{complex}$ is the change in absorbance per minute in the presence of X μM of the complex. Then, the IC_50_ values can be obtained from inhibition vs. complex concentration plots as follows: complex concentration, X [μM] = IC_50_ values where inhibition % = 50. The data can be fitted by using equation [61]:(4)$$Y(inhibition)=\frac{A\times c_{Cu\left(II\right)-complex}}{B+A\times c_{Cu\left(II\right)-complex}}$$
where *Y* stands for the inhibition, and *A* and *B* are parameters. The IC_50_ values were converted to relative activity that was calculated based on the following equation [53]:(5)$$relative\;activity=\frac{{IC}_{50}(CuZnSOD)}{{IC}_{50}(complex)}\times 100$$
and then converted to the *k_cat_* value as follows [62]:(6)$$k_{cat=}\frac{k_{NBT}\times [NBT]}{{IC}_{50}}$$

## 3. Results and Discussion

### 3.1. Protonation Equilibria of the Peptides

The deprotonation constants of the studied peptides were determined by means of pH-potentiometric titrations, and the data are collected in Table 1. All three peptides contain three or four histidine residues, while the deca- and hexadeca-peptides contain an aspartate and/or a glutamate side chain in addition to imidazole groups. The structural formulae of the studied peptides are shown in Figure 1.

The carboxyl groups in the side chains of aspartic acid and glutamic acid are deprotonated in the acid pH range, and the smallest pK value belongs to the carboxyl group of aspartic acid residue. Upon increasing the pH, the deprotonation of the imidazolium groups in the side chain of histidines occurs in overlapping steps, as indicated by the small differences in p*K*(His) values, so that the p*K* values cannot be assigned to the different imidazolium groups. However, a comparison of the pK values of L^1^(10) with those of L^2^(15) and L^3^(16) suggests that deprotonation of the C-terminal histidine (HVG(D) sequence) of the L^2^(15) and L^3^(16) peptides occurs around pH 6.5.

### 3.2. Copper(II) Complexes of the Peptides

The stoichiometry and the stability constants of the copper(II) complexes were obtained from the computer evaluation of titration curves recorded at different metal ion-to-ligand ratios. The stability constants and the derived constants are collected in Table 2. (The complexes of different ligands with the same stoichiometry have different charges; therefore, the charges are not shown.)

In the case of the 15-membered peptide, precipitation was observed in the pH range between 7 and 9, even in equimolar solution and in the presence of excess metal ions, so that the stability constants of the complexes can only be calculated with larger standard deviations.

For the studied peptides, the [CuH_x_L] species (x = 0, 1, 2) correspond to the two, three, or four imidazole coordinated monocomplexes, while the other histidine imidazole groups (in the case of x = 1, 2) are protonated.

A direct comparison between the stability constants of imidazole-coordinated species is not feasible. Therefore, derived equilibrium constants can be calculated for the complexes containing two, three, and four imidazole nitrogen atoms in the coordination sphere:(7)$$M+H_{x}L⇌{\mathrm{MH}}_{x}L$$
log *K*(M + z(N_Im_)) = log *β*(MH*_x_*L) − log *β*(H*_x_*L)(8)
where *x* is the number of the non-coordinated histidine groups. These equilibrium constants are shown in Table 2 (log *K*(M + z(N_Im_)).

The stability of the two, three, and four imidazole coordinated complexes is generally close to each other for both the peptides currently studied and the previously studied 3 and 4 histidine-containing multihistidine peptides [53,63], but the presence of one or two carboxylate groups in side chains contributes to the stability of the complex. Comparing the log*K*(Cu(II) + 3Im) values of peptides to that obtained for the L^1^(10) ligand containing two phenylalanyl side chains in addition to the glutamic carboxylate group, an extra stabilizing effect can be supposed to be due to the stacking effect between the metal ion and aromatic ring rings Thus, the stability of the three imidazole coordinated complexes increases in the order
L^2^(15) < L^3^(16) < Ac-HADHAH-NH_2_ < L^1^(10) < Ac-HDHAHDH-NH_2_,
whereas the stability of the four imidazole coordinated complexes increases in the order
L^2^(15) < L^3^(16) < Ac-HDHAHDH-NH_2_

This order also shows that the formation of macrochelate structures by coordinating histidines that are close to each other (with one or two amino acids between them) is slightly more favorable for complex formation than peptides containing histidines in a more distal position. This is consistent with previous observations [64]. Figure 1 shows the distribution curves of the complexes formed in the equimolar system of the three peptides. In the pH range indicated by the dashed line (b), a precipitate was present in the solution. These pH-potentiometric data were not taken into account for the evaluation. In all cases, the stabile three and/or four imidazole coordinated complexes are predominant species in the slightly acidic and physiological pH range (Figure 1).

However, similarly to the previously studied multihistidine peptides, the formation of imidazole-coordinated complexes cannot prevent the deprotonation and coordination of amide nitrogens of the peptide backbone. In all cases, the imidazole group of the histidine residue serves as the anchoring group to induce the deprotonation and coordination of the preceding peptide amide nitrogens. Upon increasing the pH, the deprotonation and coordination of one (two) and three amide groups take place, leading to the formation of [CuH_–1_L], [CuH_–2_L], and [CuH_–3_L] complexes. For the pentadeca- and hexadecapeptide, the deprotonation of second- and third-amide nitrogen occurs in cooperative reactions. The stepwise deprotonation constants of these processes can be found in the last four rows of Table 2.

The reported p*K*(amide) values show that stabile [CuL] complexes of peptides mimicking the binding sites of CuZnSOD enzymes shift the deprotonation of amide groups into the alkaline pH range, and the deprotonation of the first peptide nitrogen occurs at higher pH than that reported for terminally protected single histidine-containing peptides (e.g., Ac-GNGAHKPG-NH_2_ p*K*(amide) values are 6.68 (5.38) and 7.18) [65]. These effects are well reflected by the distribution curves of the copper(II) complexes (Figure 1).

The presence of three or four histidines in the peptide sequence offers more than one binding site in the molecule. As the data collected in Table 2 shows, dinuclear complexes are formed when copper(II) ion is applied in excess. In these species, similarly to the mononuclear complexes, the imidazole and 1–3 deprotonated amide nitrogen are coordinated to copper(II) ion (see Figure 2b).

Different spectroscopic methods were used to establish the coordination modes of the complexes. The visible spectral parameters of the copper(II) complexes are included in Table 3. Figure 2 represents the distribution curves of the complexes formed in the copper(II)–L^3^(16) = 2:1 system together with the change of the absorption maxima. By increasing pH, a blue shift of the absorption maxima is observed in agreement with the changes in the coordination sphere around the metal ion. The wavelengths of the measured absorption maxima are in good agreement with those obtained for other multihistidine peptides with the same coordination mode.

These data support the formation of 3 or 4 imidazole-coordinated complexes (CuL) in the physiological pH range, while 4N-coordinated species form in strong alkali medium(CuH_−3_L, Cu_2_H_−6_L). Both in equimolar solution and in Cu(II):L = 2:1 solution, an intense absorption maximum appears at around 530−540 nm in alkaline pH, confirming that the coordination mode of the metal ions in the mononuclear and dinuclear complexes is [N^-^, N^-^, N^-^, Im]. Based on the empirical formula [66], the calculated value of λ_max_ for the [N^-^, N^-^, N^-^, Im] coordination environment is expected to be 522 nm, which slightly differs from the measured values. However, we need to note that the absorption bands are quite broad, indicating the presence of coordination isomers.

Circular dichroism spectroscopic studies were carried out to confirm the assumed coordination modes of the complexes and to estimate the ratio of coordination isomers. The CD parameters of the copper(II) species are included in Table A2 in the Appendix A.

No measurable CD activity was observed in slightly acidic media, supporting the exclusive coordination of the imidazole side chains in the [CuH_x_L] and [CuL] complexes (x = 1, 2). Above pH 8, an intense positive Cotton effect is observed at around 250–300 nm, which is characteristic of the N^-^→Cu^2+^ charge transfer band. The deprotonation of the amide nitrogen atoms results in intense bands in the visible region of the spectra around 650 (negative Cotton effect) and 520 nm (positive Cotton effect). This is well presented in Figure A1 in the Appendix A, where CD spectra of the Cu(II)–L^3^(16) = 1:1 system are shown as a function of pH. Figure 3 and Figure A2 (in the Appendix A) are used to compare the spectra recorded at 1:1 and 2:1 copper(II)-to-ligand ratios.

It is obvious that CD spectra of copper(II)-L^2^(15) and copper(II)-L^3^(16) systems are similar around pH 8.0 and 9.5, and the metal ion-to-ligand ratio has no significant effect on the shape of the CD curves (Figure 3). The same similarity of the spectra can be observed at pH 11 in equimolar solutions of copper(II)-L^2^(15) and copper(II)-L^3^(16) and in the presence of copper(II) excess (Figure 3). The shape of CD spectra registered in copper(II)-L^1^(10) systems is, however, different from those of longer peptides at all studied pH. These observations can be explained by the fact that L^2^(15) and L^3^(16) ligands offer more potential binding sites for copper(II) ions, and the coordination of copper(II) ions to different histidine-containing sequences in the molecule results in different types of CD spectra. The ligands L^2^(15) and L^3^(16) contain two preferred binding sites, the N-terminal -HVH- sequence and the C-terminal -GGH- sequence. The fourth histidine is in the amino acid environment, -GPHFN-, which is less favorable for metal binding because of the presence of a proline in the vicinity of histidine. In contrast, in the L^1^(10) peptide, only the N-terminal -HVH- binding site may be the primary binding site for the metal ion. The comparison of the CD spectra of the copper(II) complexes of three oligopeptides with those of the other previously studied peptides helps to estimate the main copper(II) binding site of the ligands.

The positive Cotton effect observed at around 650 nm corresponds to the copper(II) binding site of the -HXH- sequence [63], while the band at around 500 nm can be observed when the metal ion coordinates to the -GXH- motif [65,67]. As a consequence, copper(II) ion binds mainly to the C-terminal -(NGG)HVGD- binding site in the case of 15- and 16-membered peptides. Since this part is missing from the L^1^(10) peptide, the -FHVH- part could serve as a main binding site. This is consistent with the fact that the CD spectrum of the copper(II)-L^1^(10) system is different from those of the other two peptides, and its shape is similar to that of the Ac-SarHAH-NH_2_ peptide [63]. An estimate of the ratio of the binding sites can be obtained by the superposition of two peptides modeling (Ac-SarHAH-NH_2_ [63] and Ac-PHAAA-NH_2_ [68]) the two binding sites in the appropriate ratio. Based on this, the ratio of the binding of copper(II) ion to the -FHVH- and -GPHFN- binding sites is about 60%:40% for the L^1^(10) peptide (Figure 4a). Likewise, the ratio of copper(II) ions coordinated to different metal binding sites of L^2^(15) and L^3^(16) can be estimated by the superposition of the CD spectra of Cu(II) complexes of model peptides (Figure 4b, Figure A3a in the Appendix A).

These calculations show that in equimolar solutions and alkaline media, the majority of the Cu(II) coordinates to the -GGH- motif (Figure 2a). When the metal ion is applied in two folds of excess, copper(II) ion is effectively bound to the two main binding sites, -HVH- and -GGH- sequences (Figure A3b, Figure 2b); however, the involvement of the internal histidyl residue cannot be ruled out.

Since both L^2^(15) and L^3^(16) ligands have at least three potential metal ion binding sites, both ligands can bind more than two metal ions. This is illustrated in Figure A4 (in the Appendix A), where the CD spectra are depicted at different copper(II) ion-L^3^(16) ratios. This reflects the difference between the curve recorded at 1:1, 2:1, and 3:1 metal ion-to-ligand ratios. In equimolar solutions, an intense positive Cotton effect is observed at around 500 nm, characteristic of the -GGH- sequence, indicating that copper(II) is predominantly bound by the C-terminal imidazole nitrogen in the mononuclear complexes. At a 2:1 metal ion-to-ligand ratio, a new positive band appears at around 650 nm, suggesting that one metal ion is coordinated by the N-terminal histidyl moiety while the other copper(II) is bound by the C-terminal part of the molecule.

At threefold metal ion excess, the ligand is able to bind three copper(II) ions: the negative Cotton effect at around 520 nm is characteristic of the [Im, N^-^, N^-^, N^-^] coordination mode, supporting that not only the N- and C-terminal imidazole nitrogens but also the internal histidyl residue serves as binding sites for metal ion coordination. In conclusion, the formation of trinuclear complexes is also possible in the strongly alkaline range, but pH-potentiometric measurements have not been performed to calculate their stability constants since this ratio is not relevant for biological systems, where the metal ion-to-protein ratio is much lower than 1:1.

### 3.3. Zinc(II) Complexes of the Peptides

The stability constants of the mononuclear zinc(II) complexes of the studied peptides are collected in Table 4, while the distribution curves of the complexes formed in the equimolar solution of the three ligands are shown in Figure A5 in the Appendix A. In all cases, the [ZnL] stoichiometry corresponds to the coordination via histidyl side chains. A comparison of the log*K*(Zn-3Im) values of the imidazole-coordinated species reveals that the formation constants of complexes with ligands containing aspartic and/or glutamic acid residues are higher than those without negatively charged side chains. It suggests that the interaction of the carboxylate groups enhances the stability of the imidazole-N coordinated zinc(II) complexes. Interestingly, the log*K*(Zn-3Im) value obtained for the metal complex of the L^1^(10) decapeptide containing only one negatively charged side chain is about 0.5 log unit higher than the formation constant calculated for the zinc(II)–L^3^(16) complex with tridentate imidazole-N coordination mode. It indicates that the size of the molecules and the position of the aforementioned side chains have an effect on the stability of the imidazole-coordinated complexes. This finding is in agreement with our previous results [63,64].

In slightly alkaline solutions, [ZnH_−1_L] and [ZnH_−2_L] complexes are formed, which are not strong enough to hinder the hydrolysis of the metal ion, and precipitation occurs. This suggests that these species correspond to the formation of mixed hydroxido complexes, and metal ion-induced deprotonation of amide nitrogen atoms does not take place. The same phenomenon was observed in the previously studied Zn(II)–Ac-HXHZH-NH_2_ systems [63].

### 3.4. Mixed Cu(II)/Zn(II) Complexes of the Peptides

The aforementioned results clearly reflect that in the parent copper(II) complexes of 15- and 16-membered peptides, the C-terminal histidine is the main binding site for copper(II) together with deprotonated amid nitrogen atoms, whereas for Zn(II), the main species in the physiological pH range are complexes with four imidazole coordinations. Since both metal ions bind to one of the specific peptide sequences in the native CuZnSOD enzyme, we found it worthwhile to investigate whether the binding site preference of the metal ions is altered when both metal ions are present. Thus, pH-potentiometric and spectroscopic studies of systems containing Cu(II), Zn(II) ions, and L^2^(15) or L^3^(16) peptides in equimolar concentrations were performed. In the case of the L^2^(15) peptide, only the stability constant for one complex formed in the alkaline range could be determined because precipitation occurred in the pH range between 7 and 9. The mixed metal complexes of peptide L^3^(16) could be analyzed below pH ~10 because in this case, precipitation occurred above pH 10. The precipitation redissolved only in a strongly basic solution. The stability constants of the formed mixed metal complexes are collected in Table 5.

In the mixed metal complexes, the copper(II) ion coordinates with the [(N^–^)_x_,Im] coordination mode, while the zinc(II) ion is coordinated via the N donor atom of imidazole rings. In the [CuZnH_−5_L] complex, deprotonation of one of the water molecules coordinating to Zn(II) is also assumed. Figure 5 shows the concentration distribution curves of equimolar mixed Cu(II)-Zn(II) L systems (L = L^2^(15) and L^3^(16)), where the total molar fraction of Cu(II) and Zn(II) parent complexes and mixed metal complexes are plotted, respectively.

Below pH 6, all the ligands mainly coordinate to copper(II) ion, whereas in the physiological and alkaline pH range, mixed metal complexes predominate. The formation of mixed metal complexes in the physiological and alkaline pH range can modify the distribution of metal ions between potential binding sites. CD spectroscopic studies have provided the opportunity to follow these processes. Figure 6 illustrates that in the slightly alkaline pH range, in the presence of Zn(II), the CD spectrum changes compared to the spectrum recorded in the equimolar solution of Cu(II), and the spectra of systems containing copper(II), zinc(II) and ligand in a 1:1:1 ratio are similar to those recorded at Cu(II)-L = 2:1 ratio (L = L^2^(15) and L^3^(16)). This confirms that the addition of Zn(II) results in a redistribution of Cu(II) among the potential binding sites and that Zn(II) binds at least partially to the C-terminal part of the peptides, corresponding to the Zn(II) binding site of CuZnSOD enzyme. A similar conclusion can be drawn from Figure A6 (in the Appendix), where the CD spectra of Cu(II)-Zn(II)-L^3^(16) are plotted in the function of Cu(II)-Zn(II) ratio.

### 3.5. Superoxide Dismutase Activity Measurements:

As a continuation of our work, we measured the SOD activity of the complexes of L^2^(15) and L^3^(16), and the SOD activity assay was also completed with the study of complexes of four histidine-containing heptapeptides (Ac-HDHAHDH-NH_2_). The assays were performed in samples containing equimolar amounts of Cu(II) and peptide and Cu(II), Zn(II), and peptide (*c*_L_ = *c*_Cu(II)_ = 2 μM and *c*_L_ = *c*_Cu(II)_ = *c*_Zn(II)_ = 2 µM), and the pH of the samples was adjusted to 6.8 with phosphate buffer (*c*(PO_4_^3−^) = 50 mM) as described in the experimental section. The inhibition curves recorded for Cu(II)-L^2^(15) and Cu(II)-L^3^(16) are illustrated in Figure 7. The IC_50_ values determined for all the systems studies are given in Table 6. In addition, data for previously studied multihistidine peptides and some other copper(II) and manganese(III) complexes are also included in the table. A better comparison is facilitated by the calculation of the relative activity, where the measured values are compared to the IC_50_ value of the SOD enzyme measured under the same conditions [53].

The data reflect that the SOD activity of the copper(II) complexes of the studied 15- and 16-membered peptides is outstandingly good, and similarly, good IC_50_ values were obtained for the heptapeptide containing four histidines. However, practically the same activity values were also measured in the equimolar Cu(II)-Zn(II)-L^2^(15) and Cu(II)-Zn(II)-L^3^(16) systems. Although it was expected that the rearrangement of the binding of metal ions in the presence of Zn(II) would have an effect on SOD activity, the answer is given by the distribution curves calculated for the conditions of measurement. Figure A7 and Figure A8 in the Appendix A show the distribution of Cu(II)-L^2^(15), Cu(II)-L^3^(16), and Cu(II)-Zn(II)-L^2^(15), Cu(II)-Zn(II)-L^3^(16), respectively, for the concentrations used in the SOD activity measurements. The figures clearly show that Cu(II) complexes (CuL) with only imidazole coordination dominated in the Cu(II)-L^2^(15) and Cu(II)-L^3^(16) systems under these conditions, while the proportion of Cu(II)-Zn(II) mixed complexes is much lower in Cu(II)-Zn-L (L = L^2^(15) or L^3^(16)) systems. Thus, the presence of zinc at pH 6.8 does not affect the SOD activity values. Although the relative SOD activity of the complexes studied is smaller than that of the native CuZnSOD enzyme, it is important to note that the complexes showed better performance in the degradation of superoxide anion than other SOD mimics. Thus, the incorporation of specific amino acid sequences mimicking the CuZnSOD enzyme increases the efficiency of model systems in the catalytic decomposition of superoxide anion.

## 4. Conclusions

The Cu(II), Zn(II), and Cu(II)-Zn(II) mixed complexes of 10-, 15-, and 16-membered model peptides containing both the copper(II) and zinc(II) binding sites of the CuZnSOD enzyme were investigated by pH potentiometry and spectroscopic methods, and these studies were completed by measurements of the SOD activity of Cu(II) and Cu(II)-Zn(II) mixed complexes at pH 6.8.

We have found that all the histidine imidazole rings can act as anchor groups for Cu(II). In the physiological pH range, the imidazole-coordinated complexes dominate, whereas deprotonation of amide nitrogens occurs in the alkaline pH range, and Cu(II) coordinates with the (N^-^, N^-^, N^-^, Im) donor set. In the equimolar solution, the –(NGG)HVGD– sequence of the peptides, which mimics the Zn(II) binding site of the native enzyme, is the preferred binding site, as confirmed by UV-vis and CD spectroscopic studies. However, due to the presence of more histidine, the peptides can bind more than one Cu(II) ion, and thus, in solutions containing an excess of Cu(II) ions, we observed the formation of dinuclear complexes with Cu(II) binding to the –FHVH– and –(NGG)HVGD– sequences in nearly equimolar ratios.

In Cu(II)-Zn(II)-L (L = L^2^(15) or L^3^(16)) mixed systems, the presence of Cu(II)-Zn(II) complexes could only be detected in the alkaline range. The spectroscopic measurements, however, confirm that the addition of Zn(II) results in a redistribution of Cu(II) among the potential binding site, and Zn(II) binds at least partially to the C-terminal part of the peptides, corresponding to the Zn(II) binding site of CuZnSOD enzyme.

Cu(II) complexes of 15- and 16-membered peptides showed high SOD activity. However, the presence of Zn(II) did not affect the activity because at the concentrations used in the measurements at pH 6.8, the parent Cu(II) complexes predominate.

It can be concluded that the studied model peptides, which contain the amino acid sequence corresponding to the two binding sites of the SOD enzyme, bring us closer to mimicking the enzyme. However, the effect of the larger distance between the binding sites in the native enzyme and the conformation of the molecule is not validated in these peptides, so we will continue the research with molecules obtained by further modification of these peptides.

## Data Availability

Data is contained within the article or Appendix A.

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
