# Peer review of "Characterization of Copper(II) and Zinc(II) Complexes of Peptides Mimicking the CuZnSOD Enzyme"

_molecules, 2024, doi:10.3390/molecules29040795_

Round 1
Reviewer 1 Report
Comments and Suggestions for Authors
This work deals with a very interesting scientific problem - the characterization of copper (II) and zinc (II) complexes of peptides mimicking the enzyme CuZnSOD.
The authors have chosen an adequate approach to accomplish this goal, but the goal is not clearly and precisely formulated either in the abstract or in the introduction. The introduction should justify the need for this study and conclude with a clearly stated objective.
I am sending a link that might be useful for the authors.:
https://www.scribbr.com/dissertation/abstract/
https://writing.wisc.edu/handbook/assignments/writing-an-abstract-for-your-research-paper/
Under Equations 1 and 2 on page 3, you need to define each term you have labeled.
The same applies to the following equations.
It would be good to make a separate table of the symbols and abbreviations used.
On page 6 there is something that is not at all clear what it is. Probably the formulas have been moved.
The English language in the text needs editing, I recommend that the authors consult a professional English editor.
Comments on the Quality of English LanguageThe English language in the text needs editing, I recommend that the authors consult a professional English editor.
Author Response
"Please see the attachment."

Reviewer 2 Report
Comments and Suggestions for Authors
The manuscript reports the characterization and SOD activity of three peptides, including Cu and Zn complexes. It may be of interest to individuals in the peptide science field. My comments are below.
To solidify the chemical structures of the three peptides, mass spectrometry data should be added to the manuscript.
The authors concluded in the abstract that the complexes have "significant SD activity" and in the conclusion section, "high SOD activity." However, according to Table 6 in the SOD activity measurements section, they exhibited weak SOD activity compared to CuZnSOD. I believe the conclusion needs to be reconsidered.
On pages 1 and 2, "imidazolato ring" can be replaced by "imidazole ring."
On page 1, "superoxide radical" can be replaced by "superoxide anion" because O2- is in an anion form, not radical.
Author Response
"Please see the attachment."

Reviewer 3 Report
Comments and Suggestions for Authors
The authors present measurements of peptides and spectroscopic and circular dichroism measurements at various conditions to determine the binding constants and catalytic activity of superoxide dismutase and compare the results with each other. The work is novel and definitely interesting. I suggest however to revise the manuscript according to following points prior to acceptance to reach the standard of the journal.
11. Abstract: SOD not defined upon first mentioning
22. „Cells are protected against this species by superoxide dismutase (SOD) – a metalloenzyme, which very efficiently catalyses the dismutation of superoxide radicals into H2O2 and O2. à Reference missing
33. Many antimicrobial peptides are characterised by an amphipathic structure. à For claim many 3 references missing.
44. Antimicrobial peptides include, for example, linear peptides that may adopt an α-helix conformation upon binding to bacteria. à Reference missing
55. There are peptides that form β-sheets through cysteine coupling and cysteine-constrained loop structures. à Reference missing
66. Considering the role of the secondary structure and amphiphilic nature of peptides and the composition of bacterial membranes, it was thought that the function of antimicrobial peptides involves direct binding to the lipid bilayer and that interaction with bacterial membranes is a prerequisite for the function of antimicrobial peptides. à Reference missing
77. However, the mechanism of action of antimicrobial peptides on bacteria is complex. à Reference missing
88. Parameters in equation 1 and 2 need to be defined.
99. Parameters in equation 3 and 4 need to be defined.
110. Parameters in equation 6 need to be defined.
111. Chemical equation 3: arrows are out of the equation
112. Page 8: abstract in the middle of a sentence.
113. Figure 2: Which line belongs to which axis?
114. There are other ways to prevent bacteria growth on surfaces like polymeric coating, or metal ions in the surface.1,2 The authors should mention these other approaches as competitive systems, especially, since the application of human like peptides poses threat of resistances.
115. Page 9: paragraph in the middle of a sentence
116. Figure 4: a) Error in text of the figure
117. Page 14 new abstract within sentence.
118. X-axis in Figure 7 is not scientific
References
(1) Frueh, J.; Gai, M.; Yang, Z.; He, Q. Influence of Polyelectrolyte Multilayer Coating on the Degree and Type of Biofouling in Freshwater Environment. J. Nanosci. Nanotechnol. 2014, 14 (6), 4341–4350. https://doi.org/10.1166/jnn.2014.8226.
(2) Badaraev, A. D.; Lerner, M. I.; Bakina, O. V.; Sidelev, D. V.; Tran, T.-H.; Krinitcyn, M. G.; Malashicheva, A. B.; Cherempey, E. G.; Slepchenko, G. B.; Kozelskaya, A. I.; et al. Antibacterial Activity and Cytocompatibility of Electrospun PLGA Scaffolds Surface-Modified by Pulsed DC Magnetron Co-Sputtering of Copper and Titanium. Pharmaceutics 2023, 15 (3), 939. https://doi.org/10.3390/pharmaceutics15030939.
Comments on the Quality of English Language
some wording could be updated
Author Response
"Please see the attachment."
